# Resistance to insect growth regulators and age-stage, two-sex life table in *Musca domestica* from different dairy facilities

**Naeem Abbas** *, Abdulwahab M. Hafez*

Pesticides and Environmental Toxicology Laboratory, Department of Plant Protection, College of Food and Agriculture Sciences, King Saud University, Riyadh, Saudi Arabia

* nnoor.c@ksu.edu.sa (NA); hafez@ksu.edu.sa (AMH)

**Data Availability Statement:** All relevant data are within the manuscript and its Supporting Information files.

**Funding:** The authors extend their appreciation to the Deputyship for Research & Innovation,

## Abstract

Among the vectorial insect pests, the domestic house fly (*Musca domestica* L., Diptera: Muscidae) is a ubiquitous livestock pest with the ability to develop resistance and adapt to diverse climates. Successful management of the house fly in various locations requires information about its resistance development and life table features. The status of insect growth regulators resistance and life table features on the basis of age, stage, and two sexes of the house fly from five different geographical locations of Riyadh, Saudi Arabia: Dirab, Al-Masanie, Al-Washlah, Al-Uraija and Al-Muzahmiya were therefore investigated. The range of resistance levels were 3.77–8.03-fold for methoxyfenozide, 5.50–29.75 for pyriproxyfen, 0.59–2.91-fold for cyromazine, 9.33–28.67-fold for diflubenzuron, and 1.63–8.25-fold for triflumuron in five populations of house fly compared with the susceptible strain. Analysis of life history parameters—such as survival rate, larval duration, pupal duration, pre-female duration, pre-male duration, adult and total pre-oviposition periods, longevity of male, oviposition period, female ratio, and fecundity female$^{-1}$—revealed significant variations among the field populations. Additionally, demographic features—including the generation time, the finite and intrinsic rates of increase, doubling time, and net reproductive rate—varied significantly among the field populations. These results will be helpful in planning the management of the house fly in geographically isolated dairies in Saudi Arabia.

## Introduction

*Musca domestica* L. (Diptera: Muscidae) is a ubiquitous insect pest of livestock and humans [1, 2] capable of adapting to a wide range of climates. This insect is commonly known as the house fly and serves as a carrier for many pathogens of public health and veterinary importance [3, 4]. Attempts to control the house fly with a wide range of insecticides have failed because this insect rapidly develops resistance to these chemicals [5–11]. This widespread development of insecticide resistance has necessitated the employment of integrated vector management strategies.

Various control measures are available to manage the house fly, for example, cultural practices, chemicals, and biological agents including fungal/bacterial pathogens and parasitoids/

"Ministry of Education" in Saudi Arabia for funding this research work through the project number IFKSURG-1442-480. The funders had no role in study design, data collection, and analysis, decision to publish, or preparation of the manuscript.

**Competing interests:** The authors have declared that no competing interests exist.

predators or combination of these measures [12, 13]. Insect growth regulators (IGR) comprise an effective chemical tool for integrated pest management due to their properties of high selectivity, killing immature stages, comparative environmentally safe, and reduced risk to the safety of mammals [14–16]. The juvenile hormone mimic pyriproxyfen inhibits the emergence of adults, cyromazine is a molting disrupter, diflubenzuron and triflumuron are inhibitors of chitin biosynthesis affecting chitin synthase 1, and methoxyfenozide is an ecdysone agonist [17]. These insecticides are effectively recommended to manage medical pests, including *M. domestica*, and are strong candidates for integrated pest management due to their different novel mode of actions and less environmental hazards [16, 18–20]. However, resistance to IGRs has been reported previously in house fly [20–25].

The study of life tables involves summarizing age or stage specific birth, death, and reproductive data. Classic models typically focus on females, but we will use both males and females. This methodology is a basic tool in pest management [26–28]. Comparison of the life tables of geographically separated pest populations aids in the design of effective pest management strategies. Geographic influences such as latitude and longitude and environmental influences such as humidity, photoperiod, and temperature often account for variations in the life tables of insect pests [29, 30]. Geographically and environmentally isolated pest populations have previously been reported to show variations in life tables, for instance, the house fly from urban localities in Pakistan [31]; *Parapoynx crisonalis* (Walker) (Lepidoptera: Pyralidae) [29]; *Plutella xylostella* (L.) (Lepidoptera: Plutellidae) from China [32]; and *Melanoplus femurrubrum* (De Geer) (Orthoptera: Acrididae) from USA [33]. Variations have also emerged in pesticide resistance in the house fly, *Anopheles stephensi* Liston (Diptera: Culicidae); *Oxycarenus hyalinipennis* Costa (Hemiptera: Lygaidae); *Amrasca devastans* (Distant) (Hemiptera: Cicadellidae); and *Dysdercus koenigii* (F.) (Hemiptera: Pyrrhocoridae) [5, 7, 20, 34–39].

Development of resistance in spatial dispersed pests is important to consider in resistant management strategies. For example, the populations of house fly from Punjab, Pakistan showed significant variations in resistant levels to various insecticides under different environments such as dairy (open) and poultry farms (enclosed) [5, 40]. The knowledge of insecticide resistance and life table features in the house fly from different separated dairies is very important in the design of effective management plans in different regions. Resistance to insect growth regulator insecticides and variations in the life history features of house fly from different originated dairies in Saudi Arabia are currently unexplored. Therefore, the present experiments were conducted to obtain an understanding of the resistance status and life table on the basis of age, stage and two sexes of the house fly collected from different dairies in Saudi Arabia.

## Materials and methods

### Ethics approval

No specific permit was required to collect house fly samples from the dairy farms, as these farms were privately owned and collections were made merely by personal communication with the owners.

### Collections of house fly populations

Approximately 150–200 adult house flies (of mixed sexes) were trapped in plastic jars (33 cm ×19 cm) from five dairy farms located in Riyadh, Saudi Arabia: Dirab (latitude 24.49° N, 46.62° E; elevation 576 m), Al-Masanie (latitude 24.58° N, 46.73° E; elevation 581 m), Al-Uraija (latitude 24.63° N, 46.66° E; elevation 612 m), Al-Washlah (latitude 24.54° N, 46.72° E; elevation 643 m), and Al-Muzahmiya (latitude 24.47° N, 46.27° E; elevation 764 m). These

dairies were located averagely more than 30 kilometer apart to each other. At these dairies, cultural practices (sanitation) and about 8–10 number of insecticide sprays from pyrethroids and organophosphates classes per season are being used for the control of dairy pests (Personal communication).

## Laboratory rearing of house fly populations

The rearing protocol of the house fly was adopted according to Abbas et al. [41] with some modifications. After collection, each population was shifted in a separate transparent cage (40×40 cm) in the laboratory and reared for one generation to get uniform insects. An adult diet (sugar + powdered milk at a 1:1 ratio, weighed in g) and water-soaked cotton wicks placed in 9 cm plastic petri dishes were provided for adult food. Adults were provided with fresh food every two days. Cotton wicks were moistened daily and replaced at two days interval. A paste of wheat bran, yeast, sugar and milk at a ratio of 20:5:1.5:1.5 (g), respectively, mixed with 120 mL water, was prepared in plastic cups (500 mL) and placed in adult cages for fecundity after two days rearing in the laboratory. Plastic cups with eggs were removed from adult cages daily and covered with a muslin cloth to avoid larval escape. When larvae had consumed the diet in the plastic cups, they were shifted into glass beakers with larval medium, where they became pupae. The emerged adults were shifted to cages to proceed to the next generation. All populations were maintained at 27±2°C, 65±5% humidity and a 12:12 h (L:D) photoperiod in the laboratory.

## Insecticides

Commercial-grade formulated IGR groups including pyriproxyfen (IRAC # 7C, Admiral 10EC, Sumitomo Chemicals, Japan), diflubenzuron (IRAC # 15, Diflon 250WP, Saudi Delta Company, SA), triflumuron (IRAC # 15, Starycide 480SC, Bayer Crop Sciences, Germany), cyromazine (IRAC # 17, Novasat 75WP, Astranova Chemicals, SA), and methoxyfenozide (IRAC # 18, Runner 24SC, Dow Agro-sciences, UK).

## Larval bioassays

The toxicity of IGR insecticides to the larvae of different dairy populations of *M. domestica* was determined through a diet incorporation bioassay following the method of Abbas et al. [20]. Different concentrations of each of the tested IGRs were incorporated into the larval medium (consisting of wheat bran, yeast, milk powder, and sugar at a ratio of 20:5:5:1.5:1.5 g, respectively). Five concentrations (causing mortality from >0% to <100%) of each IGR with four replicates of each were used in each bioassay. In total, 10 second instar larvae in each replicate, 40 larvae in each concentration, and 200 larvae were used in each bioassay. In the control treatment, 40 larvae with four replicates (10 larvae per replicate) were provided with larval medium without any IGR insecticide concentrations. All bioassays were conducted under the above mentioned constant conditions. Data were recorded at the emergence of adults after 3 weeks of the bioassay. Larvae that failed to transform into adults were considered as dead.

## Life table construction

To construct the life table of house fly, 100 newly (≤24 h) laid eggs were randomly collected from different egg batches at one day of each mass population and were placed in plastic cups (500 ml) supplied with artificial larval diet aforementioned above. Single egg was considered as a replicate for each population [42]. The cups were enclosed with muslin cloth to prevent larval escape. Each population was reared separately. Hatched larvae were reared in cups on

provided diet until pupation, and their developmental periods were recorded. Freshly emerged adults within 24 h were sexed and one male and one female placed into plastic jars (15 cm x 11 cm). Larval medium was provided daily in 9 cm petri dishes for oviposition. Eggs were counted daily till the death of females and egg hatchability was recorded. The female ratio, adult longevity, and oviposition period were also recorded. The experiment was conducted under the aforesaid laboratory conditions.

For each strain, the following parameters were calculated according to Chi and Su [43] and Tuan et al. [27]. Briefly, the age-specific survivorship ($l_x$) was determined as follows:

$$l_x = \sum_{j=0}^{m} s_{xj}$$

The age-stage-specific fecundity ($m_x$) was determined as follows:

$$m_x = \frac{\sum_{j=0}^{m} s_{xj} f_{xj}}{\sum_{j=0}^{m} s_{xj}}$$

The net reproductive rate (R0) was calculated as follows:

$$R0 = \sum_{x=1}^{\omega} l_x m_x$$

The intrinsic rate of increase (r) was assessed by the Lotka–Euler equation with the age indexing zero as follows:

$$\sum_{x=0}^{\infty} e^{-r(x+1)} l_x m_x = 1$$

The finite rate of increase ($\lambda$) was determined as:

$$\lambda = e^r$$

The generation time (T) was determined as follows:

$$T = \frac{\text{In}(R0)}{r}$$

The life expectancy ($e_{xj}$) was calculated as follows:

$$e_{xj} = \sum_{i=x}^{n} \sum_{j=y}^{m} s\prime_{ij}$$

The reproductive value ($v_{xj}$) was detected as follows:

$$v_{xj} = \frac{e^{r(x+1)}}{s_{xj}} \sum_{i=x}^{n} e^{-r(i+1)} \sum_{j=y}^{m} s\prime_{ij} f_{ij}$$

The gross reproductive rate (GRR) was calculated as:

$$GRR = \sum m_x$$

Where, x = alpha to beta

## Data analyses

The bioassay data were analyzed using the POLO Plus software [44] to determine the values of median lethal concentration ($LC_{50}$). In each bioassay, the mortality rates were corrected by the mortality rate obtained in the control treatment using the formula of Abbott [45], if required. Resistance ratios (RRs) were calculated as: $LC_{50}$ of field population/$LC_{50}$ of susceptible strain. The resistance levels were scaled as mention by Ahmad et al. [46]; RR = 1 (susceptibility), RR = 2–10 (low resistance), RR = 11–30 (moderate resistance), RR = 31–100 (high resistance), and RR > 100 (very high resistance).

The life table data were analyzed with the TWO-SEX-MS Chart program [47] established on the basis of two sex life table principle [28, 48]. The variances and standard errors (SE) of life history features were determined by paired bootstrap test with 100,000 replicates at $P \leq 0.05$ using the TWOSEX-MS Chart [47]. The parameters $l_x$, $s_{xj}$, $f_x$, $m_x$, $l_x m_x$, $v_{xj}$, and $e_{xj}$ were graphed with Sigma Plot 11.0.

## Results

### Resistance of *M. domestica* larvae to IGRs

The toxicity of tested IGRs was not significantly different (overlapped 95% FL) among the field populations. Low resistance against methoxyfenozide (3.77–8.03-fold) and triflumuron (1.63–8.25-fold) was detected in all the five populations of *M. domestica* in comparison to the susceptible strain. Populations collected from the Dirab, Al-Masanie, and Al-Uraija had moderate resistance (10.75–29.75-fold) to pyriproxyfen, while the Al-Washlah and Al-Muzahmiya populations had low resistance levels (5.50–8.25-fold), compared with the susceptible strain. Susceptibility to cyromazine was found in all the tested populations (0.59–1.32-fold), except the Al-Masanie population which had low resistance (2.91-fold). Populations collected from the Dirab, Al-Washlah, Al-Masanie, and Al-Muzahmiya had moderate resistance (13.67–28.67-fold) to diflubenzuron, while the Al-Uraija population had low resistance level (9.33-fold), compared with the susceptible strain (Table 1).

### Life history parameters of house flies from dairies

The larval durations of all the tested populations were significantly different from each other ($P < 0.05$). The pupal duration of the Al-Uraija population was significantly longer than that of the Dirab, Al-Masanie, Al-Washlah and Al-Muzahmiya populations ($P < 0.05$). The egg to adult duration for male flies from the Al-Masanie population was significantly longer than for flies from all other areas except Dirab ($P < 0.05$). Similarly, the egg to adult duration for female flies from the Al-Masanie population was significantly longer than for flies from all other areas ($P < 0.05$). The male total longevity in the Al-Uraija and Al-Muzahmiya populations was significantly shorter than those of the Dirab, Al-Masanie, and Al-Washlah populations ($P < 0.05$). The female total longevity in the Al-Washlah, Al-Uraija, and Al-Muzahmiya populations was significantly shorter ($P < 0.05$) than that of the Al-Masanie population (Table 2).

The total-pre-oviposition period (TPOP) and adult-pre-oviposition period (APOP) of the Al-Masanie and Al-Washlah populations were significantly longer than those of the Dirab, Al-Uraija, and Al-Muzahmiya populations ($P < 0.05$). The oviposition period were significantly reduced in the Al-Washlah population compared with that of the Al-Muzahmiya population ($P < 0.05$), but similar with the other tested populations. The female ratio in the Al-Uraija population was significantly lower compared with the Dirab and Al-Washlah populations ($P < 0.05$). Whereas the reproductive female ratio was significantly lower in the Al-Washlah population compared with the other populations ($P < 0.05$). The fecundity/female was

**Table 1. Resistance to insect growth regulators in the larvae of different dairy farms *M. domestica* populations.**

| Insecticide | Population | Year | [1]N | Slope ± SE | $\chi^2$ (df) | P | [2]$LC_{50}$ | [3]FL (95%) | [4]RR |
|---|---|---|---|---|---|---|---|---|---|
| Pyriproxyfen | Susceptible ($G_{14}$) | 2019 | 240 | 1.52 ± 0.19 | 6.17 (3) | 0.10 | 0.04 | 0.03–0.06 | 1.00 |
| | Dirab ($G_1$) | 2019 | 240 | 1.63 ± 0.27 | 5.07 (3) | 0.17 | 1.19 | 0.26–2.38 | 29.75 |
| | Al-Masanie ($G_1$) | 2019 | 240 | 2.21 ± 0.34 | 5.98 (3) | 0.11 | 0.70 | 0.16–1.19 | 17.50 |
| | Al-Uraija ($G_1$) | 2019 | 240 | 1.68 ± 0.32 | 1.52 (3) | 0.68 | 0.43 | 0.21–0.63 | 10.75 |
| | Al-Washlah ($G_1$) | 2019 | 240 | 0.80 ± 0.24 | 1.00 (3) | 0.80 | 0.22 | 0.10–0.53 | 5.50 |
| | Al-Muzahmiya ($G_1$) | 2019 | 240 | 1.31 ± 0.29 | 0.77 (3) | 0.86 | 0.33 | 0.10–0.56 | 8.25 |
| Diflubenzuron | Susceptible ($G_{14}$) | 2019 | 240 | 1.49 ± 0.34 | 2.27 (3) | 0.52 | 0.03 | 0.01–0.05 | 1.00 |
| | Dirab ($G_1$) | 2019 | 240 | 1.87 ± 0.28 | 3.75 (3) | 0.29 | 0.41 | 0.19–0.70 | 13.67 |
| | Al-Masanie ($G_1$) | 2019 | 240 | 1.68 ± 0.24 | 3.17 (3) | 0.37 | 0.46 | 0.28–0.74 | 15.33 |
| | Al-Uraija ($G_1$) | 2019 | 240 | 1.04 ± 0.24 | 0.10 (3) | 0.99 | 0.28 | 0.10–0.46 | 9.33 |
| | Al-Washlah ($G_1$) | 2019 | 240 | 1.31 ± 0.17 | 4.45 (3) | 0.22 | 0.86 | 0.55–1.40 | 28.67 |
| | Al-Muzahmiya ($G_1$) | 2019 | 240 | 1.53 ± 0.24 | 5.06 (3) | 0.17 | 0.86 | 0.37–1.78 | 28.67 |
| Triflumuron | Susceptible($G_{14}$) | 2019 | 240 | 0.99 ± 0.32 | 0.13 (3) | 0.99 | 0.08 | 0.00–0.26 | 1.00 |
| | Dirab ($G_1$) | 2019 | 240 | 2.24 ± 0.66 | 0.24 (3) | 0.97 | 0.23 | 0.04–0.39 | 2.88 |
| | Al-Masanie ($G_1$) | 2019 | 240 | 1.08 ± 0.25 | 6.05 (3) | 0.11 | 0.62 | 0.27–3.53 | 7.75 |
| | Al-Uraija ($G_1$) | 2019 | 240 | 1.83 ± 0.30 | 0.13 (3) | 0.99 | 0.66 | 0.42–0.89 | 8.25 |
| | Al-Washlah ($G_1$) | 2019 | 240 | 1.67 ± 0.38 | 0.84 (3) | 0.84 | 0.27 | 0.08–0.41 | 3.38 |
| | Al-Muzahmiya ($G_1$) | 2019 | 240 | 1.54 ± 0.47 | 1.03 (3) | 0.79 | 0.13 | 0.01–0.30 | 1.63 |
| Cyromazine | Susceptible ($G_{14}$) | 2019 | 240 | 2.54 ± 0.31 | 4.76 (3) | 0.19 | 0.22 | 0.18–0.28 | 1.00 |
| | Dirab ($G_1$) | 2019 | 240 | 2.15 ± 0.27 | 1.57 (3) | 0.67 | 0.16 | 0.13–0.21 | 0.73 |
| | Al-Masanie ($G_1$) | 2019 | 240 | 1.82 ± 0.36 | 4.76 (3) | 0.19 | 0.64 | 0.43–1.34 | 2.91 |
| | Al-Uraija ($G_1$) | 2019 | 240 | 2.76 ± 0.39 | 1.30 (3) | 0.73 | 0.29 | 0.24–0.37 | 1.32 |
| | Al-Washlah ($G_1$) | 2019 | 240 | 2.47 ± 0.32 | 6.88 (3) | 0.08 | 0.22 | 0.13–0.40 | 1.00 |
| | Al-Muzahmiya ($G_1$) | 2019 | 240 | 3.43 ± 0.39 | 6.09 (3) | 0.12 | 0.13 | 0.11–0.15 | 0.59 |
| Methoxyfenozide | Susceptible ($G_{14}$) | 2019 | 240 | 2.13 ± 0.26 | 5.38 (3) | 0.15 | 4.99 | 3.91–6.40 | 1.00 |
| | Dirab ($G_1$) | 2019 | 240 | 1.58 ± 0.36 | 3.26 (3) | 0.35 | 34.55 | 23.27–80.48 | 6.92 |
| | Al-Masanie ($G_1$) | 2019 | 240 | 3.09 ± 0.45 | 1.34 (3) | 0.72 | 40.09 | 33.46–50.03 | 8.03 |
| | Al-Uraija ($G_1$) | 2019 | 240 | 3.25 ± 0.38 | 0.94 (3) | 0.82 | 21.55 | 18.25–25.58 | 4.32 |
| | Al-Washlah ($G_1$) | 2019 | 240 | 2.15 ± 0.27 | 1.57 (3) | 0.67 | 20.79 | 16.75–26.30 | 4.17 |
| | Al-Muzahmiya ($G_1$) | 2019 | 240 | 1.56 ± 0.24 | 0.59 (3) | 0.90 | 18.79 | 14.16–25.48 | 3.77 |

[1] Number of tested larvae.

[2] Median lethal concentrations in μg/ml.

[3] Fiducial limits

[4] Resistance ratios.

significantly lower in the Al-Washlah population than that of the Al-Masanie and Al-Muzahmiya populations ($P < 0.05$), but did not differ from that of the Dirab and Al-Uraija populations (Table 2).

## Demographic life table features of house flies from dairies

The intrinsic rates of increase (r) in the Al-Masanie (0.013) and Al-Washlah (0.12) populations were significantly lower ($P < 0.05$) than that of the Dirab (0.17), Al-Uraija (0.18), and Al-Muzahmiya (0.19) populations. The finite rates of increase (λ) of the Al-Masanie and Al-Washlah (1.13) populations were significantly lower than that of the Dirab (1.18), Al-Uraija (1.19), and Al-Muzahmiya (1.21) populations ($P < 0.05$). The net reproductive rate ($R_0$) in the Al-Muzahmiya population (89.32) was significantly higher than that of the Al-Masanie (39.52)

**Table 2. Development duration and life history parameters of five house fly populations collected from dairies in Riyadh, Saudi Arabia.**

| Parameters | N | Dirab ±SEM[*] | N | Al-Masanie ±SEM[*] | N | Al-Washlah ±SEM[*] | N | Al-Uraija ±SEM[*] | N | Al-Muzahmiya ±SEM[*] |
|---|---|---|---|---|---|---|---|---|---|---|
| larvae (d) | 80 | 7.89 ± 0.05 b | 89 | 8.09 ± 0.03 a | 70 | 6.80 ± 0.11 d | 100 | 5.90 ± 0.04 e | 100 | 7.34 ± 0.11 c |
| pupae (d) | 50 | 5.68 ± 0.09 c | 27 | 6.00 ± 0.11 b | 33 | 5.58 ± 0.12 c | 70 | 6.80 ± 0.06 a | 65 | 5.11 ± 0.14 d |
| pre-adult ♂ (d) | 25 | 14.60 ± 0.14 ab | 16 | 15.00 ± 0.18 a | 13 | 14.00 ± 0.39 bc | 49 | 13.69 ± 0.11 c | 36 | 12.89 ± 0.28 d |
| pre-adult ♀ (d) | 25 | 14.56 ± 0.13 b | 11 | 15.00 ± 0.00 a | 20 | 14.35 ± 0.21 b | 21 | 13.57 ± 0.27 c | 29 | 13.35 ± 0.19 c |
| longevity ♂ (d) | 25 | 32.84 ± 2.37 b | 16 | 33.88 ± 3.56 ab | 13 | 40.46 ± 2.49 a | 49 | 22.57 ± 1.31 c | 36 | 26.28 ± 1.66 c |
| longevity ♀ (d) | 25 | 32.80 ± 2.19 ab | 11 | 39.73 ± 3.20 a | 20 | 27.00 ± 3.49 b | 21 | 27.81 ± 2.01 b | 29 | 30.52 ± 2.39 b |
| APOP[a] (d) | 19 | 5.47 ± 0.41 b | 10 | 8.70 ± 0.79 a | 9 | 8.75 ± 0.93 a | 15 | 4.80 ± 0.61 b | 20 | 4.45 ± 0.42 b |
| TPOP[b] (d) | 19 | 19.89 ± 0.42 b | 10 | 23.70 ± 0.79 a | 9 | 22.75 ± 0.58 a | 15 | 18.20 ± 0.76 c | 20 | 17.50 ± 0.45 c |
| oviposition (d) | 25 | 6.42 ± 1.12 ab | 11 | 6.90 ± 1.30 ab | 20 | 5.88 ± 1.03 b | 21 | 7.07 ± 0.70 ab | 29 | 8.95 ± 0.93 a |
| female ratio | 100 | 0.50 ± 0.007 a | 100 | 0.41 ± 0.01 ab | 100 | 0.61 ± 0.009 a | 100 | 0.30 ± 0.005 b | 100 | 0.45 ± 0.006 ab |
| rep. female ratio | 100 | 0.76 ± 0.009 a | 100 | 0.91 ± 0.009 a | 100 | 0.40 ± 0.11 b | 100 | 0.71 ± 0.10 a | 100 | 0.69 ± 0.009 a |
| fecundity/female | 25 | 298.00 ± 69.70 ab | 11 | 359.27 ± 77.79 a | 20 | 165.2 ± 52.32 b | 21 | 306.43 ± 59.48 ab | 29 | 308.00 ± 52.70 a |

Different letters in each row indicate significant differences among the populations (paired bootstrap test, $P \leq 0.05$) using TWO-SEX MS chart program.

[*] Standard errors of means (SEM) were estimated using bootstrapping (100,000 re-samplings).

N = cohort size, d = days, rep. = reproductive

[a]adult pre-oviposition period

[b]total pre-oviposition period.

and Al-Washlah (33.04) populations ($P < 0.05$), and similar to that of the Dirab (74.50) and Al-Uraija (64.35) populations. The generation time (T) in the Al-Masanie (29.43) and Al-Washlah (29.80) populations was significantly longer than that of the Dirab (26.05), Al-Uraija (23.54), and Al-Muzahmiya (24.08) populations ($P < 0.05$). The doubling time (DT) was shorter in the Al-Uraija (3.92) and Al-Muzahmiya (3.72) populations ($P < 0.05$) than that of the Al-Washlah population (5.91), but similar to that of the Dirab (4.19) and Al-Masanie (5.55) populations. Among all the tested populations, the GRR (gross reproduction rate) was similar ($P > 0.05$, Table 3).

## Age-stage-specific survival rates ($s_{xj}$) of house flies from dairies

The parameter $s_{xj}$ refers to the possible survival of newly born eggs to age x and development to stage j (Fig 1). In all the populations the peak value of $s_{xj}$ for larvae was similar. However, the highest peak values of $s_{xj}$ for pupae in the Dirab (0.80) and Al-Washlah (0.70) populations

**Table 3. Demographic life table features of five house fly populations collected from dairies in Riyadh, Saudi Arabia.**

| Parameters | Dirab ±SEM[*] | Al-Masanie ±SEM[*] | Al-Washlah ±SEM[*] | Al-Uraija ±SEM[*] | Al-Muzahmiya ±SEM[*] |
|---|---|---|---|---|---|
| intrinsic rate of increase (r) (d⁻¹) | 0.17 ± 0.01 a | 0.13 ± 0.01 b | 0.12 ± 0.02 b | 0.18 ± 0.01 a | 0.19 ± 0.01 a |
| finite rate of increase (λ) (d⁻¹) | 1.18 ± 0.01 a | 1.13 ± 0.02 b | 1.13 ± 0.02 b | 1.19 ± 0.02 a | 1.21 ± 0.01 a |
| net reproductive rate ($R_0$) (eggs/individual) | 74.50 ± 21.45 ab | 39.52 ± 13.88 b | 33.04 ± 12.15 b | 64.35 ± 17.40 ab | 89.32 ± 20.56 a |
| mean generation time (T) (d) | 26.05 ± 0.48 b | 29.43 ± 0.89 a | 29.80 ± 1.48 a | 23.54 ± 0.47 c | 24.08 ± 0.49 c |
| doubling time (DT) (d) | 4.19 ± 0.34 ab | 5.55 ± 1.07 ab | 5.91 ± 1.06 a | 3.92 ± 0.31 b | 3.72 ± 0.24 b |
| gross reproductive rate (GRR) (eggs/individual) | 305.78 ± 82.98 a | 250.15 ± 79.20 a | 240.46 ± 75.03 a | 252.04 ± 64.02 a | 344.18 ± 53.57 a |

Different letters in each row indicate significant differences among the populations (paired bootstrap test, $P \leq 0.05$) using TWO-SEX MS chart program.

[*]Standard errors of means (SEM) were estimated using bootstrapping (100,000 re-samplings). d = days.

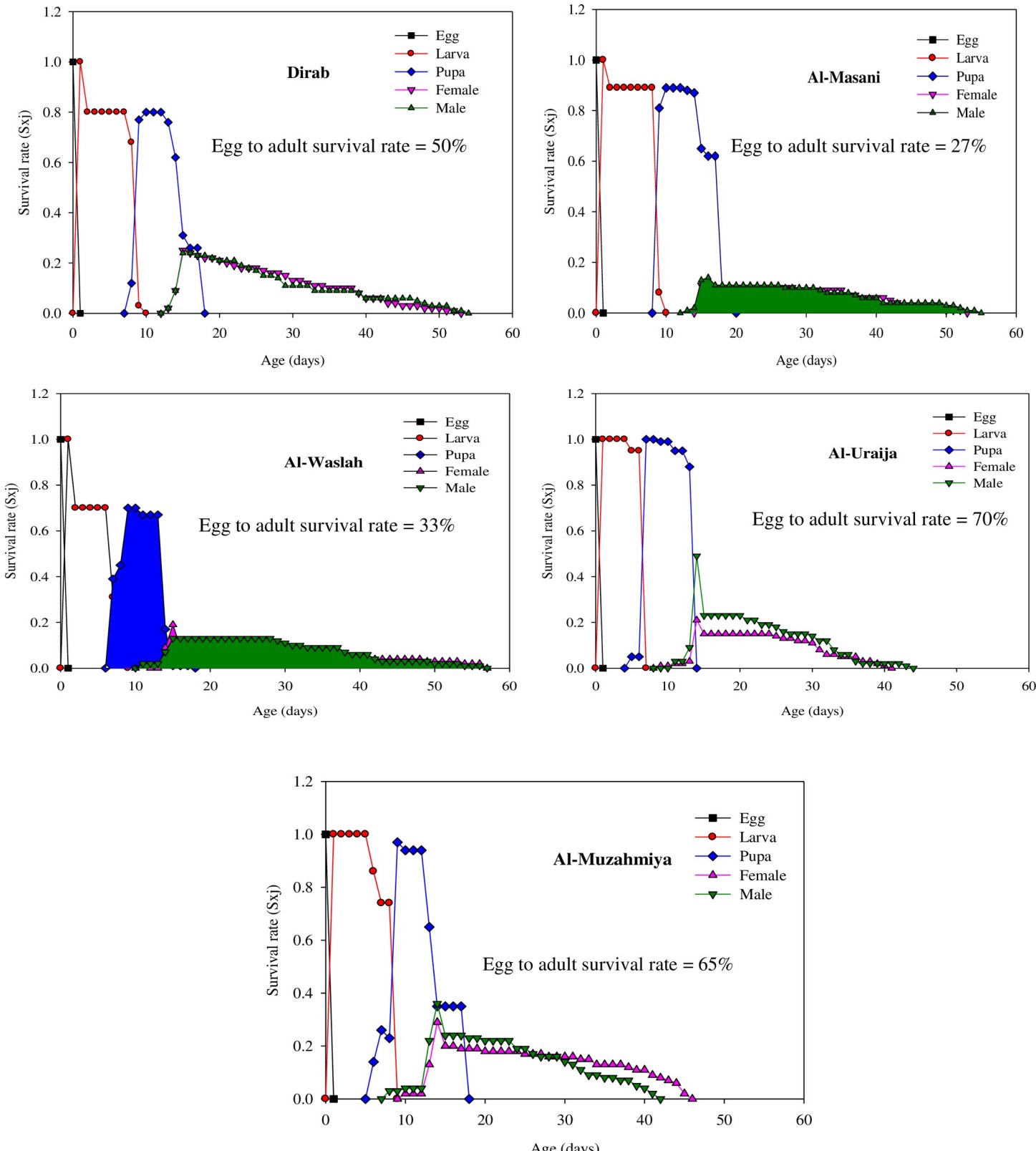

**Fig 1. Age-stage-specific survival rate ($s_{xj}$) for the Dirab, Al-Masanie, Al-Washlah, Al-Uraija, and Al-Muzahmiya populations of house fly.**

were lower than those of the Al-Masanie (0.89), Ai-Uraija (0.99), and Al-Muzahmiya (0.97) populations. Similarly, the highest peak values of $s_{xj}$ for females and males in the Dirab (0.25 and 0.24, respectively), Al-Uraija (0.21 and 0.49, respectively), and Al-Muzahmiya (0.29 and 0.36, respectively) populations were higher than those for the Al-Masanie (0.11 and 0.14, respectively) and Al-Washlah (0.19 and 0.13, respectively) populations.

### Age-specific survivorship ($l_x$), fecundities ($m_x$, $f_x$), and maternity ($l_x m_x$) of house flies from dairies

The parameters $l_x$, $f_x$, $m_x$ and $l_x m_x$ were determined for all populations (Fig 2). Among all populations, non-significant differences were found in the highest peak values for $l_x$. The highest peaks of $f_x$ in the Dirab (40.67 eggs female$^{-1}$day$^{-1}$) and Al-Muzahmiya (40.06 eggs female$^{-1}$day$^{-1}$) populations were lower than those in the Al-Masanie (44.33 eggs female$^{-1}$day$^{-1}$), Al-Washlah (43.75 eggs female$^{-1}$day$^{-1}$), and Al-Uraija (55.67 eggs female$^{-1}$day$^{-1}$) populations, while the highest peaks of $m_x$ in the Dirab (19 eggs individual$^{-1}$day$^{-1}$), Al-Masanie (20.71 eggs individual$^{-1}$day$^{-1}$), and Al-Uraija (21.97 eggs individual$^{-1}$day$^{-1}$) populations were lower than those in the Al-Washlah (25 eggs individual$^{-1}$day$^{-1}$) and Al-Muzahmiya (34.43 eggs individual$^{-1}$day$^{-1}$) populations. Additionally, the peak times of $l_x$, $f_x$ and $m_x$ for the Dirab (45th day), Al-Masanie (29th day), Al-Washlah (43th day), Al-Uraija (20th day), and Al-Muzahmiya (31th day) populations varied significantly among populations. Similarly, the highest peak values for $l_x m_x$ in the Al-Masanie (4.35 at 24th day) and Al-Washlah (2.68 at 25th day) populations were lower than those in the Dirab (6.08 at 21th day), Al-Uraija (8.35 at 20th day), and Al-Muzahmiya (7.12 at 21th day) populations.

### Age-stage-specific reproductive values ($v_{xj}$) of house flies from dairies

The $v_{xj}$ reveals the reproductive value of insects at the age x and stage j in terms of future progeny (Fig 3). The peak $v_{xj}$ values for larvae were 6.54, 3.91, 4.11, 3.44, and 5.52 day$^{-1}$ in the Dirab, Al-Masanie, Al-Washlah, Al-Uraija, and Al-Muzahmiya populations, respectively. The peak $v_{xj}$ values for pupae in the Dirab (12.35 day$^{-1}$) Al-Masanie (7.49 day$^{-1}$), Al-Uraija (11.41 day$^{-1}$), and Al-Muzahmiya (11.56 day$^{-1}$) populations were lower than those in the Al-Washlah population (18.00 day$^{-1}$). The peak $v_{xj}$ values for females in the Al-Masanie (166.7 day$^{-1}$), Al-Washlah (160.92 day$^{-1}$) and Al-Uraija (169.90 day$^{-1}$) populations were significantly higher than those in the Dirab (126.12 day$^{-1}$) and Al-Muzahmiya (132.47 day$^{-1}$) populations. The duration was shorter in the Al-Uraija and Al-Muzahmiya populations than that in the Dirab, Al-Masanie, and Al-Washlah populations.

### Age-stage-specific life expectancy ($e_{xj}$) of house flies from dairies

The $e_{xj}$ refers to the expected days over which an insect at age x and stage j will survive after age x shown in Fig 4. The $e_{xj}$ values for eggs were 22.13, 21.33, 16.48, 21.10, and 24.61 days in the Dirab, Al-Masanie, Al-Washlah, Al-Uraija, and Al-Muzahmiya populations, respectively. In the Al-Uraija and Al-Washlah populations, the peak $e_{xj}$ values for larvae were 20.1 and 20.69 days, respectively, which were shorter than those in the Dirab (25.16 days), Al-Masanie (21.72), and Al-Muzahmiya (23.61 days) populations. The peak $e_{xj}$ values for pupae were 19.16, 14.72, 17.69, 16.1, and 18.76 days in the Dirab, Al-Masanie, Al-Washlah, Al-Uraija, and Al-Muzahmiya populations, and each of these values differed significantly. The peak $e_{xj}$ values for the female and male house fly differed significantly in the tested populations—Al-Masanie (26.1 days and 24.73 days, respectively), Al-Washlah (24.88 days and 29.46 days, respectively), Al-Muzahmiya (21.60 days and 16.13 days, respectively), Dirab (19.80 days and 19.81 days, respectively), and Al-Uraija (18.80 days and 18.30 days, respectively).

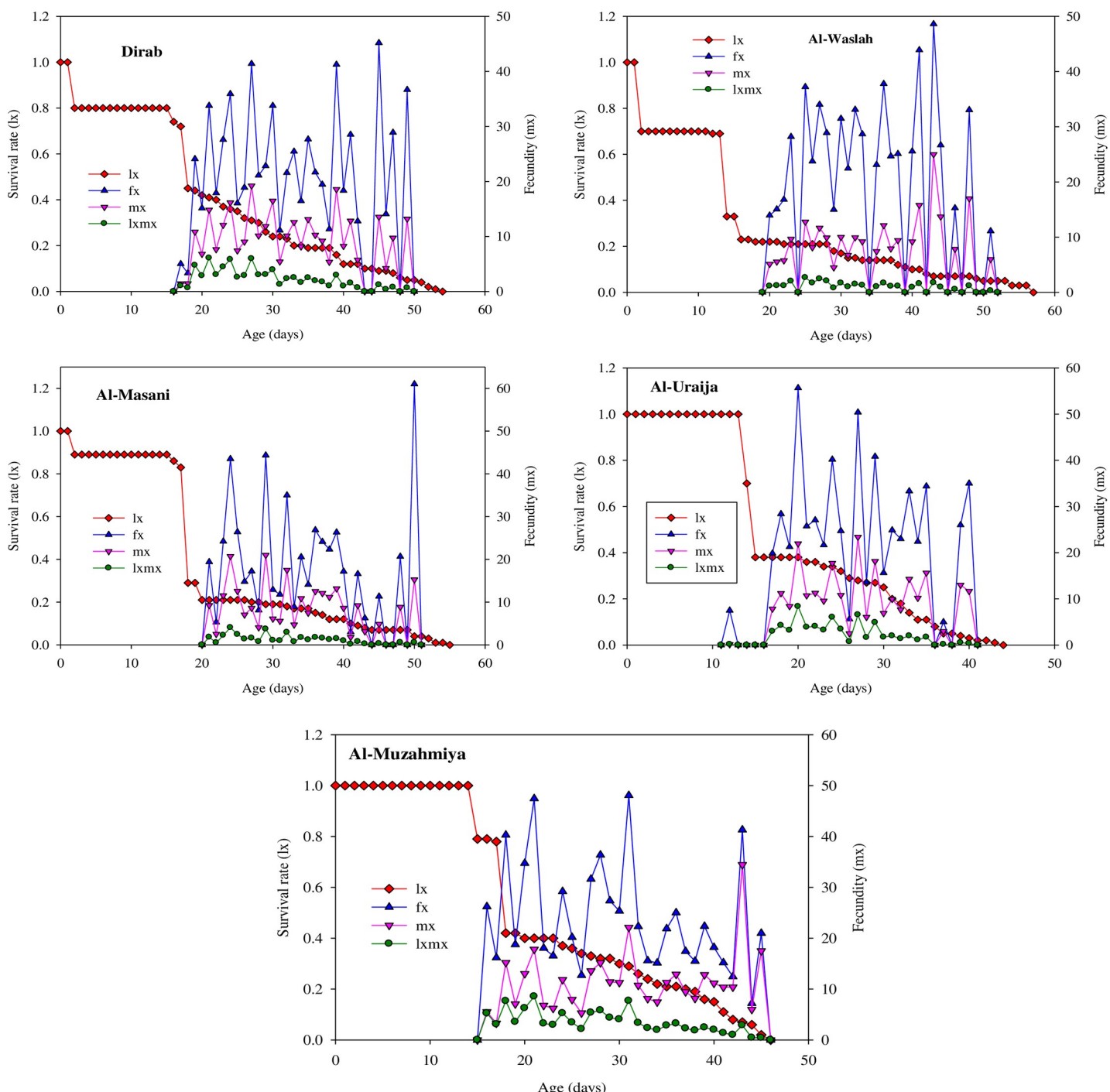

**Fig 2. Age-specific survivorship ($l_x$), age-specific female fecundity ($f_x$), age-specific fecundity of the total population ($m_x$), and age-specific maternity ($l_x m_x$) for the Dirab, Al-Masanie, Al-Washlah, Al-Uraija, and Al-Muzahmiya populations of house fly.**

## Correlation between IGR resistance ratios and life history features

There were nonsignificant correlations among the resistance ratios of tested IGRs in different house fly populations (Table 4). All the tested IGRs had nonsignificant positive and negative correlations with most of the life history features ($P > 0.05$). However, triflumuron had a

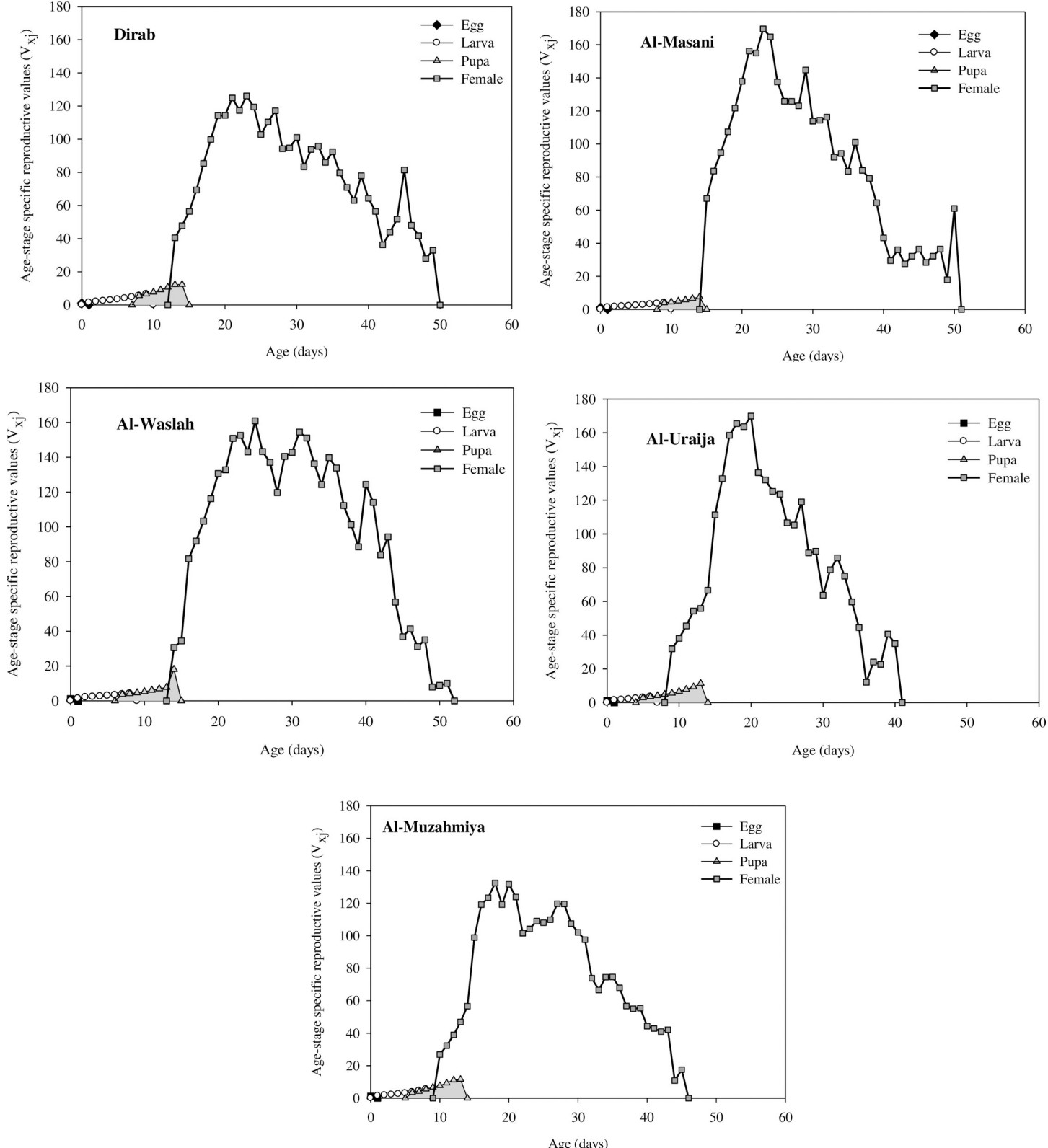

**Fig 3. Age-stage-specific reproductive values ($v_{xj}$) for the Dirab, Al-Masanie, Al-Washlah, Al-Uraija, and Al-Muzahmiya populations of house fly.**

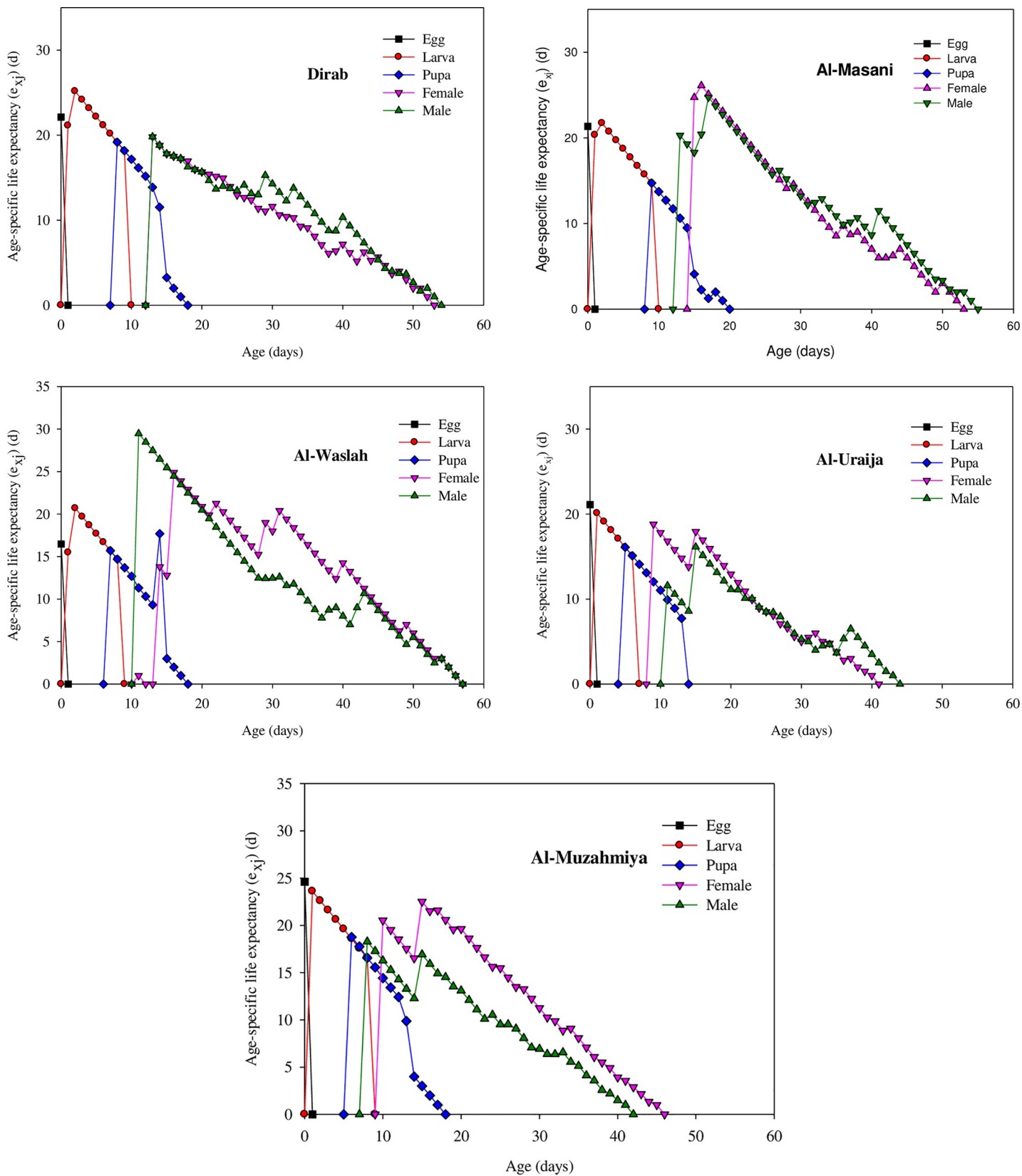

**Fig 4. Age-specific life expectancy (e$_{xj}$) for the Dirab, Al-Masanie, Al-Washlah, Al-Uraija, and Al-Muzahmiya populations of house fly.**

Table 4. Correlation of resistance ratios of insect growth regulators to the house fly populations.

| Insecticide | Pyriproxyfen | Diflubenzuron | Triflumuron | Cyromazine |
|---|---|---|---|---|
| Diflubenzuron | -0.585* | | | |
| Triflumuron | -0.007* | -0.724* | | |
| Cyromazine | 0.088* | -0.382* | 0.760* | |
| Methoxyfenozide | 0.763* | -0.511* | 0.349* | 0.681* |

*nonsignificant correlation ($P > 0.05$).

significant positive correlation with the pupal duration ($P = 0.04$). The methoxyfenozide had significant positive correlation with pre-adult male duration ($P = 0.03$) and female longevity ($P = 0.04$; Table 5).

## Discussion

In the current study, the five populations of house fly collected from dairy facilities showed susceptibility to cyromazine, low resistance to methoxyfenozide and triflumuron, low–moderate resistance to pyriproxyfen and diflubenzuron in comparison to the susceptible strain. However, the toxicity of these tested IGRs was not significantly different among the field populations. The insect populations that showed more than tenfold resistance to insecticides are known to be resistant [5, 49]. In this study, less than tenfold resistance to methoxyfenozide, cyromazine and triflumuron in all tested populations, pyriproxyfen in two populations, and diflubenzuron in one population were detected. These populations were considered tolerant rather than resistant. Previously, resistance to these insecticides has been found in house fly [20–22], *Phenacoccus solenopsis* Tinsley (Homoptera: Pseudococcidae) [50], *Spodoptera litura* (F.) (Lepidoptera: Noctuidae) [51], and *Spodoptera frugiperda* (J.E. Smith) (Lepidoptera: Noctuidae) [52]. The development of resistance depends upon the use of particular insecticides at a facility [5, 53]. In Saudi Arabia, spray applications of different insecticides are more common relative to IGRs. Therefore, similar toxicities of IGRs against house fly populations may be due to no and/or very low use of these insecticides at these dairy facilities. However, this is the first susceptibility report of house fly populations to IGRs from dairy facilities in Riyadh, Saudi

Table 5. Correlation between resistance ratios and life table features of the house fly populations.

| Life table features | Insecticides (RR) | | | | |
|---|---|---|---|---|---|
| | Pyriproxyfen | diflubenzuron | triflumuron | cyromazine | methoxyfenozide |
| larvae (d) | 0.61 | 0.10 | -0.29 | 0.31 | 0.75 |
| pupae (d) | 0.06 | -0.83 | 0.90* | 0.41 | 0.14 |
| pre-adult ♂ (d) | 0.64 | -0.50 | 0.43 | 0.67 | 0.91* |
| pre-adult ♀ (d) | 0.53 | -0.22 | 0.24 | 0.63 | 0.86 |
| longevity ♂ (d) | 0.05 | 0.43 | -0.27 | 0.15 | 0.29 |
| longevity ♀ (d) | 0.53 | -0.30 | 0.30 | 0.76 | 0.89* |
| Fecundity | 0.45 | -0.57 | 0.42 | 0.49 | 0.55 |
| intrinsic rate of increase ($d^{-1}$) | 0.13 | -0.21 | -0.21 | -0.54 | -0.36 |
| finite rate of increase ($d^{-1}$) | 0.05 | -0.08 | -0.32 | -0.63 | -0.46 |
| net reproductive rate (eggs/individual) | 0.22 | -0.03 | -0.45 | -0.61 | -0.28 |
| mean generation time (d) | -0.01 | 0.27 | 0.07 | 0.51 | 0.45 |

* Significant correlation ($P \leq 0.05$).

Arabia. Among the five tested IGRs, methoxyfenozide, cyromazine, and triflumuron were found to be the most toxic larvicides against house fly larvae. Therefore, these larvicides could be the most promising agents for the integrated vector management system in Saudi Arabia. However, the potential cross-resistance from other chemical classes [17] should be considered when aiming to design a successful IGRs rotation pattern which requires further studies to prolong the effectiveness of these larvicides. Additionally, use of cultural practices (sanitation) and biological control agents with inclusion of insecticides for the control of house fly should be considered as an integrated pest management tool [12, 13].

The adaptation of an insect pest to an altered environment and insecticide applications depends on its life table parameters, and evidence defining its population dynamics is crucial in formulating an effective pest management plan. The present study was conducted on the five house fly populations from different dairy farms in Riyadh, Saudi Arabia, to explore their life table dynamics. Results revealed prominent variations in the larval duration, pupal duration, pre-adult male and female duration, oviposition period, longevity of males and females, APOP, TPOP, female ratio, and fecundity female$^{-1}$. The reason for the differences in life history features could be attributed to altered elevations, latitudes, and environmental factors—for instance, temperature and humidity—favoring the adaptation of the house flies. Similar to those in the current results, significant variations in life history parameters have been reported in several differing isolated pest populations [29–31, 54, 55]. For example, the developmental time of *P. xylostella* populations from higher latitudes was longer than that from lower latitudes [32]. In contrast, Shirai [55] reported no difference in the immature developmental time among nine geographically separate *P. xylostella* populations. The larval duration in *Colaphellus bowringi* Baly (Coleoptera: Chrysomelidae) was increased with increasing latitudes [30], but decreased in geometrid moths; *Cabera exanthemata* (Scopoli), *Lomaspilis marginata* (L.), *Chiasmia clathrata* (L.), and *Cabera pusaria* (L.) (Lepidoptera: Geometridae) [54]. Chen et al. [29] reported a higher net reproductive rate at 24 ˚C than at other tested temperatures in *P. crisonalis*. In house fly populations from Pakistan, the immature developmental time was shorter and pupal weights were heavier in those from lower latitudes with hot climates [31]. The aforementioned variations among different insect pests could be due to the varied latitudes and favorable environments responsible for the flexibility of insect pests in their particular facilities [30, 56].

The present results exhibited variations in the demographic parameters ($r$, $\lambda$, $R_O$, T, and DT) among the five house fly populations. The $r$ and $\lambda$ of the Al-Masanie and Al-Washlah populations were significantly lower than those in other tested field populations. The Al-Muzahmiya population exhibited the highest $Ro$ among all of the field populations. The mean generation time of the Al-Uraija and Al-Muzahmiya populations was significantly shorter than that of other tested populations, while the doubling time of the Al-Washlah population was higher than those of the Al-Uraija and Al-Muzahmiya populations. The parameters $r$, $\lambda$, and $R_O$ provide the growth potential estimation of pests—a wider insight than that provided by individual life history features [2]. Because the parameters $r$ and $\lambda$ depend upon the fecundity and growth of individuals, therefore differences in these parameters might affect the expansion rates of populations [2, 57, 58]. The lower rates of increase in the populations in this study could be attributable to lower fecundity female$^{-1}$. In agreement with present results, significant variations in demographic life table parameters of pests have previously been reported in *P. crisonalis* [29], *M. femurrubrum* [33], and *P. xylostella* [32].

The parameters $s_{xj}$, $l_x$, $f_x$, $m_x$, $l_xm_x$, $v_{xj}$, and $e_{xj}$ are important indicators for evaluating the biological fitness of insect pest populations. Similar insect pests under altered environments have different survival rates, reproduction abilities, and life spans, all of which reflect specific environmental effects and species-specific parameters [59, 60]. The present results for $s_{xj}$, $l_x$, $f_x$,

$m_x$, $l_xm_x$, $v_{xj}$, and $e_{xj}$ revealed significant variations in house fly populations from five different dairy farms. Because of differing environmental factors—for instance, temperature—and extents of insecticide exposure, significant variations in these parameters have previously been documented in a range of pests, including *P. crisonalis* [29], *Sogatella furcifera* (Horvath) (Hemiptera: Delphacidae) [60], *Bradysia odoriphaga* Yang and Zhang (Diptera: Sciaridae) [42], *Phthorimaea operculella* (Zeller) (Lepidoptera: Gelechiidae) [61], *S. litura* [27], and *Tetranychus urticae* Koch (Acari: Tetranychidae) [62]. The house fly has 5–7 kilometer flight range and dispersal [63], whereas in the present study, the house fly populations were collected from averagely more than 30 kilometer apart located dairies. The differences in life history features may be due to different populations and differences in extent of management activities for house fly in specific dairy facilities by the dairy owners.

Effectiveness and implementation of control measures against insect pests depend on knowledge of the life table attributes of any pest in its respective environment [31, 32]. In the present study, *M. domestica* populations collected from Dirab, Al-Uraija, and Al-Muzahmiya facilities showed faster development and better life table parameters than those from the Al-Masanie and Al-Washlah facilities. Rapid growth of house fly in these dairy facilities may enhance the insecticidal usage, which ultimately may lead to resistance problem in the future. Therefore, the management of house fly should be focused on specific dairy facility. Moreover, nonsignificant correlations between life history features and IGRs suggest that the integration of IGRs with cultural practices in these dairy facilities may suppress the house fly growth and resistance problem. Our results provide useful insights about the age, stage, and two sexes based life table variations in house fly populations from Riyadh, Saudi Arabia for the better management in specific dairy facilities.

## Supporting information

**S1 File. Life table of *Musca domestica* population collected from Dirab, Saudi Arabia.** (XLSX)

**S2 File. Life table of *Musca domestica* population collected from Al-Masanie, Saudi Arabia.** (XLSX)

**S3 File. Life table of *Musca domestica* population collected from Al-Washlah, Saudi Arabia.** (XLSX)

**S4 File. Life table of *Musca domestica* population collected from Al-Uraija, Saudi Arabia.** (XLSX)

**S5 File. Life table of *Musca domestica* population collected from Al-Muzahmiya, Saudi Arabia.** (XLSX)

## Acknowledgments

The authors thank the Deanship of Scientific Research and Researchers Support and Services Unit (RSSU) at King Saud University for their technical support. Authors would like also to thank the researchers and technicians from Pesticides and Environmental Toxicology Laboratory: Ahmed Mohammed Dabo and Safwat Gamal Sabra for their help in collecting and maintaining the house fly field populations in the laboratory and for other laboratory work.

## Author Contributions

**Conceptualization:** Naeem Abbas, Abdulwahab M. Hafez.

**Formal analysis:** Naeem Abbas, Abdulwahab M. Hafez.

**Investigation:** Naeem Abbas.

**Methodology:** Naeem Abbas.

**Resources:** Abdulwahab M. Hafez.

**Supervision:** Abdulwahab M. Hafez.

**Validation:** Abdulwahab M. Hafez.

**Visualization:** Naeem Abbas.

**Writing – original draft:** Naeem Abbas.

**Writing – review & editing:** Naeem Abbas, Abdulwahab M. Hafez.

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
