## [Decision Letter · Decision Letter 0]

4 Nov 2020

PONE-D-20-31395

Resistance to insect growth regulator insecticides and age-stage, two-sex life table parameters in house fly, Musca domestica from different dairy facilities

PLOS ONE

Dear Dr. Abbas,

Thank you for submitting your manuscript to PLOS ONE. After careful consideration, we feel that it has merit but does not fully meet PLOS ONE’s publication criteria as it currently stands. Therefore, we invite you to submit a revised version of the manuscript that addresses the points raised during the review process.

We look forward to receiving your revised manuscript.

Kind regards,

Ahmed Ibrahim Hasaballah

Academic Editor

PLOS ONE

Journal Requirements:

2. In your Methods section, please provide additional information regarding the permits you obtained for the work. Please ensure you have included the full name of the authority that approved the collection sites access and, if no permits were required, a brief statement explaining why.

Additional Editor Comments:

The article is interesting and seems useful in managment of housefly on a worldwide scale. The methods and data seem basically sound. However, the article still needs some modifications to increase readability. There is a mismatch between text and tables where the sample size in the tables exceeds that expected based on the text. In addition, language improving is needed.

Reviewers' comments:

Reviewer's Responses to Questions

**Comments to the Author**

1. Is the manuscript technically sound, and do the data support the conclusions?

Reviewer #1: Yes

Reviewer #2: Yes

2. Has the statistical analysis been performed appropriately and rigorously? 

Reviewer #1: Yes

Reviewer #2: Yes

3. Have the authors made all data underlying the findings in their manuscript fully available?

Reviewer #1: No

Reviewer #2: Yes

4. Is the manuscript presented in an intelligible fashion and written in standard English?

Reviewer #1: No

Reviewer #2: Yes

5. Review Comments to the Author

Reviewer #1: This is a review of the paper entitled “Resistance to insect growth regulator insecticides and age-stage, two-sex life table parameters in house fly, Musca domestica from different dairy facilities.” The research looked at resistance development for five insecticides in the house fly at five dairy farms in Saudi Arabia. The research is useful in documenting the current state of resistance.

1) English could be improved. It was clear enough that I felt I understood the manuscript, but there were significant problems.

2) Expand thoughts into sentences. Line 50 is an example.

Original: “The study of life tables on the theory of age, stage, and two sexes of insect pests …”

Problems: Studying life tables is the start of a scientific discipline called demography. Studying the tables without underlying theory is of little use. There is no theory of “age, stage, and two sexes.”

One solution: Demography is a basic and important tool in pest management [22-24].

Another solution) The study of life tables involves summarizing age or stage specific birth, death, and reproductive data. Classic models typically focus on females, but we will use both males and females. This methodology is a basic tool in pest management [22-24].

3) It would help to put the insecticides into their IRAC classification: pyriproxyfen (7C), triflumuron and diflubenzuron (15), cyromazine (17), and methoxyfenozide (18).

4) Maybe put the insecticides in order. The order could be alphabetical, or in order of increasing/decreasing activity, or by IRAC classification. Alphabetical is my least favorite. Assertions later in the manuscript suggest ordering by elevation or latitude/longitude.

5) The authors calculate all the variables that they can find, but it is difficult to grasp an overall sense of what these differences mean. Would the authors consider a final figure that shows the estimated population growth over several generations? That way we can see how early reproduction versus many offspring plays out over time. If it makes sense the model could be driven by a degree-day approach to further customize outcomes to environmental conditions at each dairy.

6) Tables and text MUST agree. The text (Lines 99-110) state that an assay consisted of 5-concentrations, 4-replicates, 10-larvae = 200. The control had 4-replicates of 10-larvae = 40. N in table 1 has 240, or 280 for treatments, and 210, or 270 for the controls.

7) The data analysis is incomplete. Tables 2 and 3 have letters typically used in some type of multiple comparison procedure. The table footnote indicates a “paired bootstrap test” but no further details are given. Does the test correct for the experimentwise error rate, or is it a bootstrap version of a t-test that should have had a Bonferroni correction applied?

8) Is there any record of current practices at these dairies? What they have used in the past, application frequency, that sort of thing?

Specific comments:

Line 34) “capable to adapt” change to “capable of adapting to”

Line 34) “This insect is”

Line 34) “as the house”

Line 35) change serve to serves

Line 36) Attempts to control the house fly with a wide range of insecticides has failed because this insect rapidly develops resistance to these chemicals.

Line 41-42) “comparatively environmentally safe” is a sentence fragment.

Line 50) “The study of life tables on the theory of age, …”

Line 64) depends

Line 64) If true then livestock/urban pest control professionals need to read more of the resistance management literature from agriculture where mode of action rotations, and other methods have been developed. “Environmental factors” as resistance management tools probably would not even make the list of most tree and row crops producers. However, the difference between open versus closed systems is a good point. Closed systems are becoming more popular in the agricultural sector whether entirely artificial or screened enclosures.

Line 78) I would remove “after attraction by a mixture of dried milk and sugar” and make this text a new sentence. As written, I might think that the milk and sugar came from five dairy farms.

Line 84, 101, 123, and elsewhere) (37] Why a mix of round and square brackets?

Line 86) This is an unanswerable question with these data, but how quickly is resistance lost under zero-stress laboratory conditions? If there is a competitive disadvantage to resistance in the absence of insecticide then the reported values are some lower bound to the true value.

Line 88) delete “after”

Line 89) delete “with a micro-syringe (5 ml)” In this case the issue is that the use of a micro-syringe is unimportant detail and it is unclear if the syringe volume was 5 ml or you used the syringe to deliver 5 ml to the wick. In either case the volume of water needed will depend on the number of flies, temperature, and humidity in the cage.

Paragraph line 99) How was this put together? I could run 1 cup at each concentration every week, or I could make a large batch of diet and fill all 4 cups of each concentration at once. Were the larvae of same age or different ages? Were the larvae still alive? What fraction were alive versus dead?

Line 112) “100 freshly laid eggs at different days were randomly…” This sentence is confusing. I will guess that you found freshly laid (define fresh) egg masses and arbitrarily removed one egg per egg mass. This was done over several days until you had 100 individuals.

Line 115) 3 does not go into 100 evenly.

Line 117) Freshly emerged adults within 24 h were sexed and one male and one female placed into plastic jars (15 cm x 11 cm).

121) You do not conduct parameters under laboratory conditions.

Line 140) The equations are meaningless without indicating what all the symbols mean. Some definitions are in the tables.

Line 127) follows not follow

Line 127) Ro is also sometimes Ro, R0 or are these all different?

Equations) There are odd mistakes in the equations. If A=b and b=1 then does not A=1? The equation for T is simply wrong. Use parentheses as appropriate to make things clearer. For example, sqrtx should not be written when you mean sqrt(x). “sqrtx” is a variable name, while “sqrt(x)” indicates applying sqrt to a variable “x”. As written I need a definition for “In” and I need a definition for “Ro” and I am missing an operator telling me what to do with these (add, subtract, multiple, divide, something else..).

Line 149) The web link for citation 42 did not work. It returns “can’t reach this page” error.

Line 147) do not use tolerance. Tolerance has a specific definition in host plant resistance. In this context it is confusing. If the insect tolerates the insecticide is it not resistant? Low-level resistance is clear enough as it is unclear if low-level resistance is significantly different from susceptible. In some cases yes (methoxyfenozide) and other cases sometimes. For pyriproxyfen no for Al0Washlah, but yes for Al-Muzahmiya based on overlap of the 95% FL.

Line 156) Do not use tolerance.

Line 167) “different from” not “different to”

Line 169) What is pre-male? Do you mean male larvae? I assume that you sex the adults and work backwards.

Line 169-170) “The pre-male duration of the Al-Masanie” is more clearly written as “The egg to adult duration for male flies from the Al-Masanie population was significantly longer than for flies from all other areas except Dirab.”

Line 171) pre-female same problem as with pre-male.

Line 177) Avoid acronyms whenever possible. A) In a table where space is limited use acronyms and define in footnotes. B) TPOP for total pre-oviposition period is a good acronym if you need to use it dozens of times. If the goal is to make it easier for readers to find relevant parts in tables/figures then define at each use in text: total pre-oviposition period (TPOP) …

Table 1) The locations are in different order for each insecticide. Why?

Table 1) Why are the chi-square degrees of freedom sometime 3, 4, or 6?

Table 1) Why is N sometimes 210, 240, 270, or 280?

Table 2) For Al-Washlah: N=8 for TPOP yet N=20 for oviposition. I have magically gained 12 individuals. There are similar problems elsewhere.

Figure 1) No clue why some area is clear other colored.

Figure 1) The figures look more like an age/stage decomposition of survival where “death” in one stage can be mortality or “birth” into the next stage. However, “birth” rate in this sense is not exactly survival rate.

That said, the figures are already hard to read and another set of points will make that problem worse. The individual graphs have considerable white space. Could a small inset of “total survival” be added?

Figure 3) Filled areas obscure parts of the graph.

Line 212) “house flies from dairies” is a better choice than dairy house flies because it is clearer that these are house flies that happen to be collected at dairies.

Line 262) “found” rather than “occurred”

Line 269) That is a major point in the insecticide resistance action committee (IRAC) work. https://irac-online.org/

Line 312) Does the range in latitude/longitude in this study warrant such attribution relative to the range in the cited studies? If the study sites are arranged by latitude/longitude would they support this hypothesis? This is a large enough piece of work that you do not need to speculate on unfounded hypotheses. Besides, if the real answer is latitude/longitude then there is nothing that these farmers can do to solve their problem.

Line 312) How would you separate the effect of latitude/longitude from geographic distance? The dairies that are 2-kilometers distant will have more in common than ones that are 2000-km distant.

Line 312) if the dairies are ordered by elevation does a pattern emerge? You have five sites and five estimates of r, does a linear regression show a relationship between r and elevation in your data? If not, then this is pointless speculation that can be deleted.

Reviewer #2: Review report:

Manuscript Number: PONE-D-20-31395

Full Title: Resistance to insect growth regulator insecticides and age-stage, two-sex life table

parameters in house fly, Musca domestica from different dairy facilities

Comments:

The topic of the article is interesting and bears implications in management of housefly (Musca domestica) on a worldwide scale. Although the present study considered the housefly from dairy facilities, still, the information is useful for the population regulation of housefly elsewhere where the insecticide resistance is a problem. The study design including the experiments and statistical analyses and the compilation appears to be suitable for addressing the issues of insecticide resistance of housefly. However, few minor corrections are required with reference to the language and the title of the article.

Suggestions:

1] Please consider a concise title of this article.

2]The hypothesis of your study should be mentioned clearly at the end of the introduction section.

3]Please use an alternative to life table traits…you can consider life table features.

4]Please check for the typographical errors if any.

5] Please consider the following articles for inclusion in discussion section:

Kočišová A, Petrovský M, Toporčák J, Novák P. 2004. The potential of some insect growth regulators in housefly (Musca domestica) control. Biologia, 59/5: 661—668.

(Please consider this article as important to compare and highlight the impact of the insect growth regulators in housefly regulation. Please consider inclusion of the content of this paper in the introduction section.)

Malik A, Singh N, Satya S. 2007. House fly (Musca domestica): A review of control strategies for a challenging pest. Journal of Environmental Science and Health, Part B, 42:4, 453-469. [Please consider this article as important to highlight the various control measures available….both in the introduction and discussion sections]

Farooq M, Freed S. 2016. Infectivity of housefly, Musca domestica (Diptera: Muscidae) to different entomopathogenic fungi. Brazilian Journal of Microbiology, 47: 807 -816.

[ Please consider this article for highlighting the fungal agents in biocontrol of housefly]

Abbas N, & Khan A, Shad, S. 2014. Resistance of the house fly Musca domestica (Diptera: Muscidae) to lambda-cyhalothrin: Mode of inheritance, realized heritability, and cross-resistance to other insecticides. Ecotoxicology (London, England). 23. 10.1007/s10646-014-1217-7. [Please consider this article for the elaboration of resistance to pesticides by the housefly]

Please improve the quality of the figures, if possible. particularly Figure 3

6. PLOS authors have the option to publish the peer review history of their article (what does this mean?). If published, this will include your full peer review and any attached files.

Reviewer #1: No

Reviewer #2: No

---

## [Author Response · Author response to Decision Letter 0]

5 Jan 2021

PONE-D-20-31395

Resistance to insect growth regulator insecticides and age-stage, two-sex life table parameters in house fly, Musca domestica from different dairy facilities

Answer: Carefully checked for style

2. In your Methods section, please provide additional information regarding the permits you obtained for the work. Please ensure you have included the full name of the authority that approved the collection sites access and, if no permits were required, a brief statement explaining why.

Answer: Ethic statement added in the manuscript

Additional Editor Comments:

The article is interesting and seems useful in managment of housefly on a worldwide scale. The methods and data seem basically sound. However, the article still needs some modifications to increase readability. There is a mismatch between text and tables where the sample size in the tables exceeds that expected based on the text. In addition, language improving is needed.

Answer: Corrected and English language carefully checked and improved. The manuscript is edited for English language and spelling by Enago, an editing brand of Crimson Interactive Inc.

Reviewer #1: This is a review of the paper entitled “Resistance to insect growth regulator insecticides and age-stage, two-sex life table parameters in house fly, Musca domestica from different dairy facilities.” The research looked at resistance development for five insecticides in the house fly at five dairy farms in Saudi Arabia. The research is useful in documenting the current state of resistance.

1) English could be improved. It was clear enough that I felt I understood the manuscript, but there were significant problems.

Answer: Thanks for comment, Carefully checked and improved

2) Expand thoughts into sentences. Line 50 is an example.

Original: “The study of life tables on the theory of age, stage, and two sexes of insect pests …”

Problems: Studying life tables is the start of a scientific discipline called demography. Studying the tables without underlying theory is of little use. There is no theory of “age, stage, and two sexes.”

One solution: Demography is a basic and important tool in pest management [22-24].

Another solution) The study of life tables involves summarizing age or stage specific birth, death, and reproductive data. Classic models typically focus on females, but we will use both males and females. This methodology is a basic tool in pest management [22-24].

Answer: Done as suggested

3) It would help to put the insecticides into their IRAC classification: pyriproxyfen (7C), triflumuron and diflubenzuron (15), cyromazine (17), and methoxyfenozide (18).

Answer: Done as suggested

4) Maybe put the insecticides in order. The order could be alphabetical, or in order of increasing/decreasing activity, or by IRAC classification. Alphabetical is my least favorite. Assertions later in the manuscript suggest ordering by elevation or latitude/longitude.

Answer: Done by IRAC classification

5) The authors calculate all the variables that they can find, but it is difficult to grasp an overall sense of what these differences mean. Would the authors consider a final figure that shows the estimated population growth over several generations? That way we can see how early reproduction versus many offspring plays out over time. If it makes sense the model could be driven by a degree-day approach to further customize outcomes to environmental conditions at each dairy.

Answer: Thanks for comment, unfortunately we did not take the data of population growth for several generations, already a big set of data is presented in the manuscript but we will consider in future experiments.

6) Tables and text MUST agree. The text (Lines 99-110) state that an assay consisted of 5-concentrations, 4-replicates, 10-larvae = 200. The control had 4-replicates of 10-larvae = 40. N in table 1 has 240, or 280 for treatments, and 210, or 270 for the controls.

Answer: Corrected and cleared

7) The data analysis is incomplete. Tables 2 and 3 have letters typically used in some type of multiple comparison procedure. The table footnote indicates a “paired bootstrap test” but no further details are given. Does the test correct for the experiment wise error rate, or is it a bootstrap version of a t-test that should have had a Bonferroni correction applied?

Answer: Data were analysed by paired bootstrap test using TWO-SEX MS chart program. Lettering is done manually based on confidence intervals difference and P value (<0.05). Standard errors of means (SEM) were estimated using bootstrapping (100,000 re-samplings) using TWOSEX MS chart program. 

8) Is there any record of current practices at these dairies? What they have used in the past, application frequency, that sort of thing?

Answer: Added in material and method section

Specific comments:

Line 34) “capable to adapt” change to “capable of adapting to”

Answer: Done as suggested

Line 34) “This insect is”

Answer: Done as suggested

Line 34) “as the house”

Answer: Done as suggested

Line 35) change serve to serves

Answer: Done as suggested

Line 36) Attempts to control the house fly with a wide range of insecticides has failed because this insect rapidly develops resistance to these chemicals.

Answer: Done as suggested

Line 41-42) “comparatively environmentally safe” is a sentence fragment.

Answer: Rephrased

Line 50) “The study of life tables on the theory of age, …”

Answer: Rephrased

Line 64) depends

Answer: Done as suggested

Line 64) If true then livestock/urban pest control professionals need to read more of the resistance management literature from agriculture where mode of action rotations, and other methods have been developed. “Environmental factors” as resistance management tools probably would not even make the list of most tree and row crops producers. However, the difference between open versus closed systems is a good point. Closed systems are becoming more popular in the agricultural sector whether entirely artificial or screened enclosures.

Answer: Rephrased

Line 78) I would remove “after attraction by a mixture of dried milk and sugar” and make this text a new sentence. As written, I might think that the milk and sugar came from five dairy farms.

Answer: Removed as suggested

Line 84, 101, 123, and elsewhere) (37] Why a mix of round and square brackets?

Answer: Corrected

Line 86) This is an unanswerable question with these data, but how quickly is resistance lost under zero-stress laboratory conditions? If there is a competitive disadvantage to resistance in the absence of insecticide then the reported values are some lower bound to the true value.

Answer: Human error, the populations were reared for one generation, corrected

Line 88) delete “after”

Answer: Done

Line 89) delete “with a micro-syringe (5 ml)” In this case the issue is that the use of a micro-syringe is unimportant detail and it is unclear if the syringe volume was 5 ml or you used the syringe to deliver 5 ml to the wick. In either case the volume of water needed will depend on the number of flies, temperature, and humidity in the cage.

Answer: Deleted

Paragraph line 99) How was this put together? I could run 1 cup at each concentration every week, or I could make a large batch of diet and fill all 4 cups of each concentration at once. Were the larvae of same age or different ages? Were the larvae still alive? What fraction were alive versus dead?

Answer: Bioassay for each insecticide and each population were conducted separately. Fresh solution was made for every population. Same age (2nd instar) larvae were used. Diet was mixed with a concentration for 4 replications together and then equally provided in four cups. Alive larvae were fed on the treated diet till pupation. Data was recorded on adult emergence. We did not noted fraction in larval stage, we noted emerged adults, from this mortality was calculated, For example, for one concentration 40 larvae were exposed, out of this 4 adults emerged, it means 36 larvae were died for that concentration and so on… 

Line 112) “100 freshly laid eggs at different days were randomly…” This sentence is confusing. I will guess that you found freshly laid (define fresh) egg masses and arbitrarily removed one egg per egg mass. This was done over several days until you had 100 individuals.

Answer: Rephrased

Line 115) 3 does not go into 100 evenly.

Answer: Rephrased

Line 117) Freshly emerged adults within 24 h were sexed and one male and one female placed into plastic jars (15 cm x 11 cm).

Answer: Done

121) You do not conduct parameters under laboratory conditions.

Answer: Corrected

Line 140) The equations are meaningless without indicating what all the symbols mean. Some definitions are in the tables.

Answer: Added

Line 127) follows not follow

Answer: Done

Line 127) Ro is also sometimes Ro, R0 or are these all different?

Equations) There are odd mistakes in the equations. If A=b and b=1 then does not A=1? The equation for T is simply wrong. Use parentheses as appropriate to make things clearer. For example, sqrtx should not be written when you mean sqrt(x). “sqrtx” is a variable name, while “sqrt(x)” indicates applying sqrt to a variable “x”. As written I need a definition for “In” and I need a definition for “Ro” and I am missing an operator telling me what to do with these (add, subtract, multiple, divide, something else..).

Answer: Ro similar, Corrected

Line 149) The web link for citation 42 did not work. It returns “can’t reach this page” error.

Answer: New version available at: http://140.120.197.173/Ecology/prod02.htm

Line 147) do not use tolerance. Tolerance has a specific definition in host plant resistance. In this context it is confusing. If the insect tolerates the insecticide is it not resistant? Low-level resistance is clear enough as it is unclear if low-level resistance is significantly different from susceptible. In some cases yes (methoxyfenozide) and other cases sometimes. For pyriproxyfen no for Al0Washlah, but yes for Al-Muzahmiya based on overlap of the 95% FL.

Answer: Deleted

Line 156) Do not use tolerance.

Answer: Deleted

Line 167) “different from” not “different to”

Answer: Done

Line 169) What is pre-male? Do you mean male larvae? I assume that you sex the adults and work backwards.

Answer: egg to adult duration for male, changed

Line 169-170) “The pre-male duration of the Al-Masanie” is more clearly written as “The egg to adult duration for male flies from the Al-Masanie population was significantly longer than for flies from all other areas except Dirab.”

Answer: Done as suggested

Line 171) pre-female same problem as with pre-male.

Answer: Changed as suggested for male

Line 177) Avoid acronyms whenever possible. A) In a table where space is limited use acronyms and define in footnotes. B) TPOP for total pre-oviposition period is a good acronym if you need to use it dozens of times. If the goal is to make it easier for readers to find relevant parts in tables/figures then define at each use in text: total pre-oviposition period (TPOP) …

Answer: Change as suggested

Table 1) The locations are in different order for each insecticide. Why?

Answer: Changed and made consistent for each insecticide

Table 1) Why are the chi-square degrees of freedom sometime 3, 4, or 6?

Answer: Corrected

Table 1) Why is N sometimes 210, 240, 270, or 280?

Answer: Human error, corrected

Table 2) For Al-Washlah: N=8 for TPOP yet N=20 for oviposition. I have magically gained 12 individuals. There are similar problems elsewhere.

Answer: Corrected, total females were 20, and 9 females produced eggs

Figure 1) No clue why some area is clear other colored.

Answer: Corrected

Figure 1) The figures look more like an age/stage decomposition of survival where “death” in one stage can be mortality or “birth” into the next stage. However, “birth” rate in this sense is not exactly survival rate. That said, the figures are already hard to read and another set of points will make that problem worse. The individual graphs have considerable white space. Could a small inset of “total survival” be added?

Answer: Added

Figure 3) Filled areas obscure parts of the graph.

Answer: Corrected

Line 212) “house flies from dairies” is a better choice than dairy house flies because it is clearer that these are house flies that happen to be collected at dairies.

Answer: Changed

Line 262) “found” rather than “occurred”

Answer: Done

Line 269) That is a major point in the insecticide resistance action committee (IRAC) work. https://irac-online.org/

Answer: Added

Line 312) Does the range in latitude/longitude in this study warrant such attribution relative to the range in the cited studies? If the study sites are arranged by latitude/longitude would they support this hypothesis? This is a large enough piece of work that you do not need to speculate on unfounded hypotheses. Besides, if the real answer is latitude/longitude then there is nothing that these farmers can do to solve their problem.

Answer: Deleted

Line 312) How would you separate the effect of latitude/longitude from geographic distance? The dairies that are 2-kilometers distant will have more in common than ones that are 2000-km distant.

Line 312) if the dairies are ordered by elevation does a pattern emerge? You have five sites and five estimates of r, does a linear regression show a relationship between r and elevation in your data? If not, then this is pointless speculation that can be deleted.

Answer: Deleted

Review report: 

Manuscript Number: PONE-D-20-31395

Full Title: Resistance to insect growth regulator insecticides and age-stage, two-sex life table

parameters in house fly, Musca domestica from different dairy facilities

Comments:

The topic of the article is interesting and bears implications in management of housefly (Musca domestica) on a worldwide scale. Although the present study considered the housefly from dairy facilities, still, the information is useful for the population regulation of housefly elsewhere where the insecticide resistance is a problem. The study design including the experiments and statistical analyses and the compilation appears to be suitable for addressing the issues of insecticide resistance of housefly. However, few minor corrections are required with reference to the language and the title of the article.

Suggestions:

1] Please consider a concise title of this article. 

Answer: Done

2]The hypothesis of your study should be mentioned clearly at the end of the introduction section.

Answer: Done

3]Please use an alternative to life table traits…you can consider life table features.

Answer: Done

4]Please check for the typographical errors if any.

Answer: Checked and removed

5] Please consider the following articles for inclusion in discussion section:

Kočišová A, Petrovský M, Toporčák J, Novák P. 2004. The potential of some insect growth regulators in housefly (Musca domestica) control. Biologia, 59/5: 661—668.

(Please consider this article as important to compare and highlight the impact of the insect growth regulators in housefly regulation. Please consider inclusion of the content of this paper in the introduction section.) 

Malik A, Singh N, Satya S. 2007. House fly (Musca domestica): A review of control strategies for a challenging pest. Journal of Environmental Science and Health, Part B, 42:4, 453-469. [Please consider this article as important to highlight the various control measures available….both in the introduction and discussion sections] 

Farooq M, Freed S. 2016. Infectivity of housefly, Musca domestica (Diptera: Muscidae) to different entomopathogenic fungi. Brazilian Journal of Microbiology, 47: 807 -816.

[ Please consider this article for highlighting the fungal agents in biocontrol of housefly]

Abbas N, & Khan A, Shad, S. 2014. Resistance of the house fly Musca domestica (Diptera: Muscidae) to lambda-cyhalothrin: Mode of inheritance, realized heritability, and cross-resistance to other insecticides. Ecotoxicology (London, England). 23. 10.1007/s10646-014-1217-7. [Please consider this article for the elaboration of resistance to pesticides by the housefly]

Answer: Added above references as suggested

6] Please improve the quality of the figures, particularly, Figure 3.

Answer: Improved

---

## [Decision Letter · Decision Letter 1]

1 Feb 2021

PONE-D-20-31395R1

Resistance to insect growth regulators and age-stage, two-sex life table in  Musca domestica from different dairy facilities

PLOS ONE

Dear Dr. Abbas,

Thank you for submitting your manuscript to PLOS ONE. After careful consideration, we feel that it has merit but does not fully meet PLOS ONE’s publication criteria as it currently stands. Therefore, we invite you to submit a revised version of the manuscript that addresses the points raised during the review process.

We look forward to receiving your revised manuscript.

Kind regards,

Ahmed Ibrahim Hasaballah

Academic Editor

PLOS ONE

Reviewers' comments:

Reviewer's Responses to Questions

**Comments to the Author**

1. If the authors have adequately addressed your comments raised in a previous round of review and you feel that this manuscript is now acceptable for publication, you may indicate that here to bypass the “Comments to the Author” section, enter your conflict of interest statement in the “Confidential to Editor” section, and submit your "Accept" recommendation.

Reviewer #2: All comments have been addressed

Reviewer #3: All comments have been addressed

2. Is the manuscript technically sound, and do the data support the conclusions?

Reviewer #2: Yes

Reviewer #3: No

3. Has the statistical analysis been performed appropriately and rigorously? 

Reviewer #2: Yes

Reviewer #3: No

4. Have the authors made all data underlying the findings in their manuscript fully available?

Reviewer #2: Yes

Reviewer #3: Yes

5. Is the manuscript presented in an intelligible fashion and written in standard English?

Reviewer #2: Yes

Reviewer #3: Yes

6. Review Comments to the Author

Reviewer #2: The authors have addressed the questions raised on the earlier version of the manuscript. The manuscript may be accepted for publication.

Reviewer #3: 29 January 2021

General comments:

Although the authors have made several corrections based on the reviewers’ comments, there are still several concerns that I have mentioned both here and in the attached file. However, these amendments do not alleviate my fundamental concerns about the experimental design, validation, interpretation, and presentation of results.

This article deals with the investigation of resistance in five populations of houseflies to five IGR insecticides. The authors then try to explain the reasons behind this resistance to IGRs through life table studies. There are several main concerns that authors need to answer:

How can authors prove that houseflies collected from each area are a separate population? As the authors have presented in the methods, the coordinates of the sampling sites including geographical coordinates and altitude, this difference of locations cannot prove a significant effect on the fact that each is a separate population. With these coordinates, I cannot imagine a distinctive feature for each geographical area as a separate population. Also, houseflies can move easily between different areas in many ways.

The authors obtained the median lethal concentration (LC50) and resistance ratio (RR) for each IGR for each region and listed them in the table, but did not compare the correlation between LC50 values and RR of five IGRs in one region relative to the other regions. Or they have not examined the resistance of five regions to one IGR with another IGRs. In this manuscript, the authors provide only a simple descriptive report of IGR toxicity among areas. Therefore, it is not clear how the relationship of houseflies to IGRs in the regions are, and that based on this experiment it has not been answered whether each region can be considered a separate population or not. I became curious and calculated these correlational relationships, and as I expected, the populations' reaction to the IGRs was significantly correlated, and this result proves that they might be the same population.

In the following, the authors have only examined the life table of houseflies in each area. But they only reported the results of this study and no scientific interpretation has been provided behind it. It seems that the authors tried to interpret the reasons for the differences among the regions to the IGRs with the help of life table parameters, but did not achieve this goal. Although they have given a detailed report of the life table parameters in the results, the big issue with the discussion is they didn't really provide logical interpretation and validation of the results. Here the authors must examine the correlations between the main parameters of the life table with the LC50 and the RR, and provide a scientific, reasoned, and logical interpretation. The authors tried to state that the parameters of the life table have significant differences in different regions. It is usual to expect this because these differences are not so far from expectation. But that was not the purpose of the study at all.

Specific comments:

- line 24 and others: oviposition period

- line 46: give the full name of CHSI when first mentioned.

-line 60 and others: replace “L.” with “Linnaeus”. Keep uniformity.

Line 64: Replace F. with Fabricius.

Line 101 and others: replace mL with ml.

Line 131: newly laid eggs, do you mean £24 h??

Line 143 and others: use the word “survivorship” for lx. Use the word “survival rate” for Sxj.

Line 143-161: there are many errors in the formula of life table parameters that I have corrected in the attached file.

Line 377, 403, 407, 410, 413, 449, 453, 455, 476, 479, 513, and 544: Journal names should be written in the abbreviation that I have written the correct acronym for the journal in the attached file.

7. PLOS authors have the option to publish the peer review history of their article (what does this mean?). If published, this will include your full peer review and any attached files.

Reviewer #2: No

Reviewer #3: No

---

## [Author Response · Author response to Decision Letter 1]

2 Mar 2021

Dear editor,

Thanks for valuable comments; we have incorporated all the comments in the manuscript.

General comments:

Although the authors have made several corrections based on the reviewers’ comments, there are still several concerns that I have mentioned both here and in the attached file. However, these amendments do not alleviate my fundamental concerns about the experimental design, validation, interpretation, and presentation of results. 

This article deals with the investigation of resistance in five populations of houseflies to five IGR insecticides. The authors then try to explain the reasons behind this resistance to IGRs through life table studies. There are several main concerns that authors need to answer:

How can authors prove that house flies collected from each area are a separate population? As the authors have presented in the methods, the coordinates of the sampling sites including geographical coordinates and altitude, this difference of locations cannot prove a significant effect on the fact that each is a separate population. With these coordinates, I cannot imagine a distinctive feature for each geographical area as a separate population. Also, houseflies can move easily between different areas in many ways.

The authors obtained the median lethal concentration (LC50) and resistance ratio (RR) for each IGR for each region and listed them in the table, but did not compare the correlation between LC50 values and RR of five IGRs in one region relative to the other regions. Or they have not examined the resistance of five regions to one IGR with another IGRs. In this manuscript, the authors provide only a simple descriptive report of IGR toxicity among areas. Therefore, it is not clear how the relationship of houseflies to IGRs in the regions are, and that based on this experiment it has not been answered whether each region can be considered a separate population or not. I became curious and calculated these correlational relationships, and as I expected, the populations' reaction to the IGRs was significantly correlated, and this result proves that they might be the same population.

Answer: The flight range of house fly is 5-7 km (Nazni et al. 2005). 

“Nazni et al. (2005) Determination of the flight range and dispersal of the house fly, Musca domestica (L.) using mark release recapture technique. Tropical Biomedicine 22(1):53-61”

In our study, the selected dairy facilities were averagely more than 30 km away from each other. Moreover, the populations were collected in a same day, so there is no chance of population drift. So these populations are different, but they showed similar toxicity to insect growth regulators. This may be due to no or little use of IGRs at these facilities, because mostly spray applications of pyrethroids and organophosphates have been done in dairy farms (Personal communication with dairy owners). We provided the correlation between RR of five IGRs in all regions (Table 4).

Similarly, there are many studies published in which similar toxicity of different IGRs in different field populations of house fly is reported.

1) Survey of insect growth regulator (IGR) resistance in house flies (Musca domestica L.) from southwestern Turkey. Journal of Vector Ecology 34 (2): 329-337. 2009.

2) Resistance Status of Musca domestica L. Populations to Neonicotinoids and Insect Growth Regulators in Pakistan Poultry Facilities. Pakistan J. Zool., vol. 47(6), pp. 1663-1671, 2015.

3) Toxicity and resistance of field collected Musca domestica (Diptera: Muscidae) against insect growth regulator insecticides. Parasitol Res (2016) 115:1385–1390

In the following, the authors have only examined the life table of houseflies in each area. But they only reported the results of this study and no scientific interpretation has been provided behind it. It seems that the authors tried to interpret the reasons for the differences among the regions to the IGRs with the help of life table parameters, but did not achieve this goal. Although they have given a detailed report of the life table parameters in the results, the big issue with the discussion is they didn't really provide logical interpretation and validation of the results. Here the authors must examine the correlations between the main parameters of the life table with the LC50 and the RR, and provide a scientific, reasoned, and logical interpretation. The authors tried to state that the parameters of the life table have significant differences in different regions. It is usual to expect this because these differences are not so far from expectation. But that was not the purpose of the study at all.

Answer:

The parameters of the life table have significant differences in different regions, it is clear cut indication that how much significant the study was and importance of the results to provides pivotal information for the house fly management relative to different locations. We provided the correlations between the main parameters of the life table with the RR (Table 5). Also we improved the discussion and provided the reasons and interpretation of the results.

Specific comments:

- line 24 and others: oviposition period

Answer: Done

- line 46: give the full name of CHSI when first mentioned.

Answer: Done

-line 60 and others: replace “L.” with “Linnaeus”. Keep uniformity.

Answer: Done

Line 64: Replace F. with Fabricius.

Answer: Done

Line 101 and others: replace mL with ml.

Answer: Done

Line 131: newly laid eggs, do you mean £24 h??

Answer: Done

Line 143 and others: use the word “survivorship” for lx. Use the word “survival rate” for Sxj.

Answer: Done

Line 143-161: there are many errors in the formula of life table parameters that I have corrected in the attached file. 

Answer: Done

Line 377, 403, 407, 410, 413, 449, 453, 455, 476, 479, 513, and 544: Journal names should be written in the abbreviation that I have written the correct acronym for the journal in the attached file.

Answer: Done

---

## [Editor Report · Decision Letter 2]

4 Mar 2021

Resistance to insect growth regulators and age-stage, two-sex life table in  Musca domestica from different dairy facilities

PONE-D-20-31395R2

Dear Dr. Abbas,

We’re pleased to inform you that your manuscript has been judged scientifically suitable for publication and will be formally accepted for publication once it meets all outstanding technical requirements.

Kind regards,

Ahmed Ibrahim Hasaballah

Academic Editor

PLOS ONE

---

## [Editor Report · Acceptance letter]

31 Mar 2021

PONE-D-20-31395R2 

Resistance to insect growth regulators and age-stage, two-sex life table in *Musca domestica* from different dairy facilities 

Dear Dr. Abbas:

I'm pleased to inform you that your manuscript has been deemed suitable for publication in PLOS ONE. Congratulations! Your manuscript is now with our production department. 

Kind regards, 

on behalf of

Dr. Ahmed Ibrahim Hasaballah 

Academic Editor

PLOS ONE